# Experimental Investigation of Clearance Influences on Cage Motion and Wear in Ball Bearings

**Baogang Wen** [1], **Meiling Wang** [2], **Xu Zhang** [1,*], **Jingyu Zhai** [3] and **Wei Sun** [3]

1 School of Mechanical Engineering and Automation, Dalian Polytechnic University, Dalian116034, China; wbg_dlut@163.com
2 College of Locomotive and Rolling Stock Engineering, Dalian Jiaotong University, Dalian 116028, China; meilingcc@163.com
3 School of Mechanical Engineering, Dalian University of Technology, Dalian 116024, China; zhaijy@dlut.edu.cn (J.Z.); sw_dlut@163.com (W.S.)
* Correspondence: zhangxu@dlpu.edu.cn; Tel.: +86-411-8632-3682

**Abstract:** Clearances of cages in ball bearings, including pocket and guiding clearances, play a vital role in the stability and reliability of bearings. In this paper, experiments on the cage motion and wear were carried out to investigate the influence of clearances in ball bearings. Firstly, the cages with a series of pocket and guiding clearances were specially designed and tested for prescribed operating conditions on a bearing test rig in which the cage motions were measured, and corresponding wear was also observed. Then, the normalized trajectory, waveform, and spectra of cage motion were constructed and compared to illustrate the effects of clearances on the cage motion and then to establish the relationship between cage motion and wear. Results reveal that the cage motion and wear are both significantly affected by its clearances. The increment of cage guiding clearance makes the whirl trajectories of the cage regular and the motion frequency of cage motion significantly change. However, the increment of cage pocket clearance make the whirl trajectories change from well-defined patterns to complicated ones, and the frequency of cage motion apparently changes. Additionally, the bearing wear is closely related to the cage motion. If the inner ring frequency is of domination for the cage motion, the cage guiding surface will wear seriously. While cage motion is dominated by two times cage frequency in spectrum domain, the cage pocket will wear more seriously.

**Keywords:** cage motion; cage wear; guiding clearance; pocket clearance; characteristic frequency

## 1. Introduction

A cage is a key component in rolling bearings to ensure stable running [1]. The cage clearance is one of the important parameters influences on bearing properties, such as lubrication [2], oil film [3], and heating [4], and especially cage motion and instability [5]. Meanwhile, the cage motion plays a decisive role in the dynamic performance and service life of the bearing, even leading to abnormal operation of the bearing and occasionally premature bearing failure [6,7]. Thus, the investigation of cage clearance on its complicated motions and wear is of great importance for the design, optimization, and evaluation of cages and has become an important research topic.

Some theoretical investigations of cage motion affected by cage clearances have been carried out by scholars. Ghaisas [8] and Liu [9] discussed the effects of the ratio of pocket clearance to guiding clearance on the cage motion in a cylindrical roller bearing and ball bearing, respectively, and indicated that cage motion stability is strongly dependent on the roller-cage pocket clearance. Further, the cage motion are evaluated as a function of clearances both in the ball pockets and at the guiding lands by Gupta [5,10], with the well-known computer code ADORE (Advanced Dynamics Of Rolling Elements) and demonstrated that cage instability has a definite correlation to cage pocket and guiding

land clearances. Meeks [11] analyzed the cage motion and forces affected by varying cage clearances and found that if a bearing has a larger ball pocket clearance compared with the guiding clearance, this may result in cage nonsynchronous motion and lower ball forces.

Besides the theoretical literature above, several experimental studies of bearing properties with respect to cage clearances have also been conducted. Adams [12] indicated that the geometry of the cage was a major contributor to the instability of the bearings. Ryu [13] experimentally investigated the cage instability with respect to ball-pocket clearances by measuring the friction coefficient of bearing and sound vibration and found larger ball-pocket clearance to be more stable. Sathyan [14] compared the frictional torque under circular pocket to square pocket retainers and pointed out that the circular pocket showed low frictional torque and was more prone to instability compared to the square pocket. As for the cage motion measurements in rolling bearings, Gupta [15] checked a certain shaft speed at which the cage can run into a whirling pattern. Sakaguchi [16] experimentally investigated the whirl amplitudes of the cage in a tapered roller bearing affected by inner-ring rotational speed and axial load. Wen et al. [17] and Li et al. [18] experimentally investigated the rotating speeds and external loads on the cage motion in a ball bearing. Chen et al. [19] experimental investigated the dynamic motions of a cage under various external radial loads during acceleration and deceleration for an angular ball bearing. Choe et al. [20] experimentally studied the dynamic behavior of ball caused by cage guidance and pocket clearances in cryogenic environments.

Although some studies on cage motion, as well as on how frictional torque and sound are affected by cage clearances or geometry, have been carried out, experimental investigations concerning the cage motion affected by its clearances and the relationship between cage motion and wear are also inadequate and unknown at present. In this paper, cage motion and the relationship between motion and wear in a ball bearing with different clearances were investigated experimentally. Firstly, a bearing test rig was specially developed to measure the spatial motions of the cage, and then six diverse pocket and guiding clearances of the cage were designed and carried out on the test rig at prescriptive operating conditions. Additionally, different patterns of cage motion and wear were then obtained and compared based on the results to investigate the influence of the cage clearances and the relationship between cage motion and wear. All the experimental results are useful references for cage ball-pocket clearance design and optimization.

## 2. Experimental Approach

A bearing test rig was developed to measure the motions and wear, in which the cage was specifically redesigned with diverse clearances including the pocket and guiding ones. The pocket clearance is the radius differences between cage pocket and ball, can be expressed as $C_p = 0.5(D_p - D_b)$, as shown in Figure 1. Additionally, guiding clearance as the radius differences between the two guiding surfaces of cage and inner ring can be expressed as $C_g = 0.5(D_{ci} - D_{ig})$, as shown in Figure 1, where $D_p$, $D_b$, $D_{ci}$, $D_{ig}$, $B_g$ are the diameters of cage pocket and ball, the inner diameter of the cage, outer diameter of inner ring, and the width of cage, respectively. As a certain bearing, different clearances can be obtained by changing pocket and inner diameter of the cage.

### 2.1. Description of Test Rig

The specially established bearing test rig is shown in Figure 2, which consists of motor, coupling, supports, shaft, and bearing box. The tested bearing of 7013AC with redesigned cage was installed on the end of the cantilever shaft, while the inner ring was rotating with the shaft. The bearing was tested under combined loads, where a constant radial force (along the *z* direction) and axial force (along the *x* direction) were applied on the bearing house and the corresponding values were measured by force transducers as shown in Figure 2b. The outer ring was cut specially to make the eddy transducer probes closer to the cage surface by a processing method of wire electrode discharge machining (WEDM). Two probes ($y_c$, $z_c$) were installed in a bearing house 90 degrees apart and obtained with

a data acquisition system to measure the radial cage motion. Additionally, the measured cage motion was analyzed by fast Fourier transform (FFT) and trajectories. More details for the test rig can be found in Ref. [17].

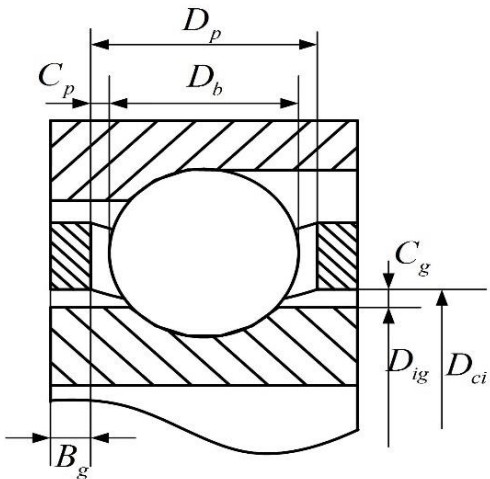

**Figure 1.** Structural diagram of cage clearances.

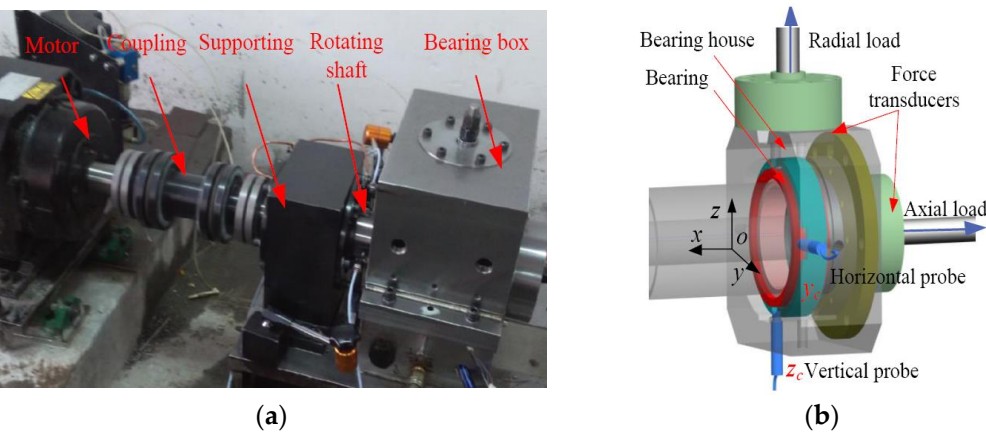

**Figure 2.** Bearing test rig. (**a**) Photo of test rig, (**b**) internal structure of bearing box.

### 2.2. Tested Bearing and Cage Design

The angular contact ball bearing of 7013AC was chosen in this paper for experimental investigation. The main geometrical parameters are listed in Table 1.

**Table 1.** Geometrical parameters of an angular contact ball bearing 7013AC.

| Parameter | Value | Unit |
|---|---|---|
| Ball diameter | 11.1 | mm |
| Bearing outside diameter | 100 | mm |
| Inner ring diameter | 65 | mm |
| Initial contact angle | 25 | deg |
| Inner curvature factor | 0.515 | - |
| Outer curvature factor | 0.51 | - |
| Pitch diameter | 82.525 | mm |
| Cage outside diameter | 86.5 | mm |
| Cage width | 26 | mm |
| Number of balls | 18 | - |

In order to investigate the cage motion affected by the cage guiding clearance, three cages with constant pocket clearance but different guiding clearances were designed with the cage inner diameter $D_{ci}$ of 75.5 mm, 75.7 mm, and 76.1 mm as shown in Figure 3. Correspondingly, $C_g$ was 0.1 mm, 0.2 mm, and 0.4 mm, respectively, and the constant pocket clearance was set as 0.1 mm. For the influence studies about of the pocket clearance, three cages with diverse pocket clearances (0.1 mm, 0.25 mm, and 0.4 mm) and constant guiding clearance of 0.45 mm are designed, in which pocket diameter $D_p$ was 11.3 mm, 11.6 mm, and 11.9 mm, respectively, as shown in Figure 4. In addition, the material of the cage is 45# steel.

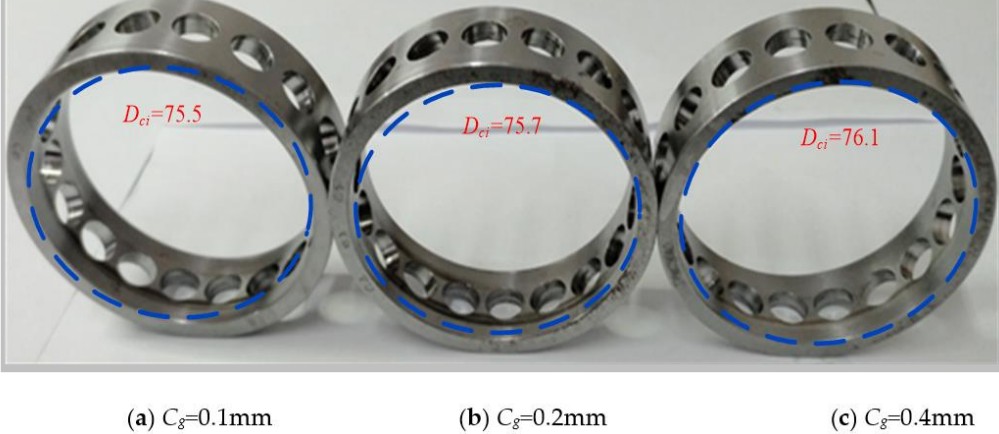

(**a**) $C_g$=0.1mm          (**b**) $C_g$=0.2mm          (**c**) $C_g$=0.4mm

**Figure 3.** Cage with different guiding clearances.

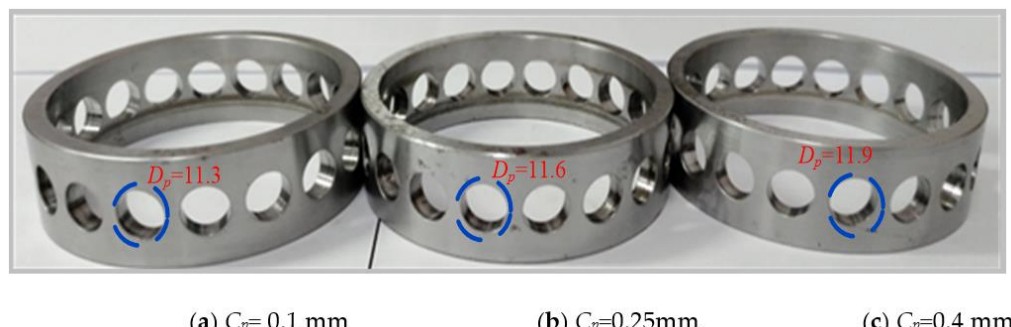

(**a**) $C_p$= 0.1 mm          (**b**) $C_p$=0.25mm          (**c**) $C_p$=0.4 mm

**Figure 4.** Cage with different pocket clearances.

Oil-injection lubrication with flow speed of 0.1 L/min was adopted for the tested bearing. The axial load and radial load applied on the tested bearing could be from 50 N to 2000 N, and operating speed could range from 600 to 6000 rpm. A variety of experiments with different loads and operating speeds were carried out for further study.

## 3. Effect of Cage Guiding Clearance

### 3.1. Radial Motions of the Bearing Cage

In order to find out the effect of the law of clearance, the cage motion had to be normalized by the corresponding clearances. The vibration waveforms, spectra of the cage motion ($z_c$) in vertical direction, and the normalizations of trajectories of the cage mass center for the tested bearing with constant preload (axial load 1000 N and radial load 500 N) and three guiding clearances at a rotating speed 6000 rpm are illustrated in Figure 5.

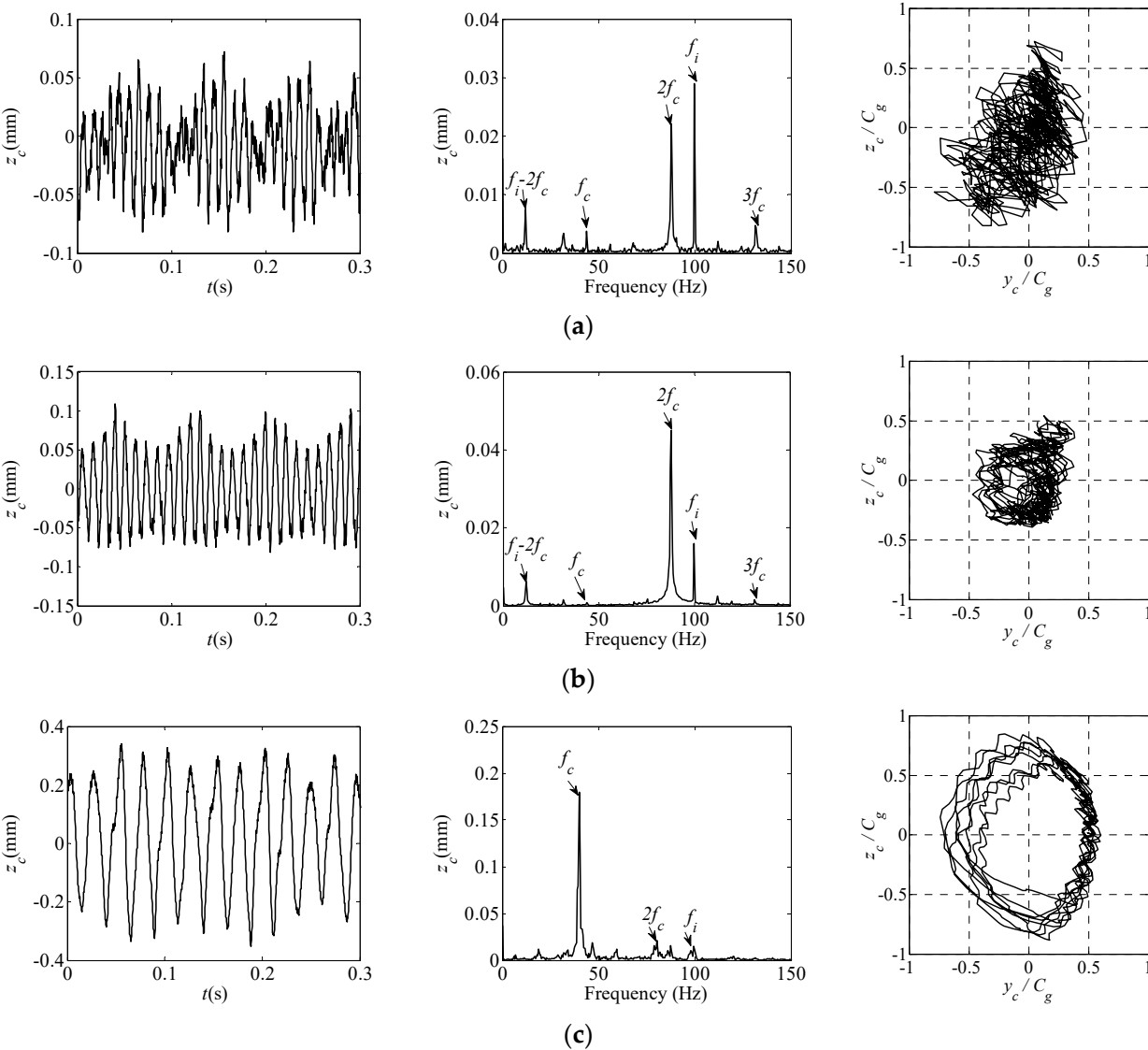

**Figure 5.** Radial motions of the cage with different guiding clearances ($\omega_i$ = 6000 rpm). (**a**) Waveform, spectra and normalized trajectory of cage motion when $C_g$ = 0.1 mm; (**b**) waveform, spectra and normalized trajectory of cage motion when $C_g$ = 0.2 mm; (**c**) waveform, spectra and normalized trajectory of cage motion when $C_g$ = 0.4 mm.

The cage rotating frequency $f_c$ can be expressed as follows:

$$f_c = \frac{f_i(1 - D_b \cos \alpha^0 / d_m)}{2} \tag{1}$$

where $f_i = \omega_i/60$ is the rotating frequency of the inner ring (Hz), $\omega_i$ is the inner ring rotating speed, $\alpha^0$ is the original contact angle, and $d_m$ is the bearing pitch diameter.

As shown in Figure 5, the motions of the cage are obviously periodic with the cage rotating frequency. Its two times, i.e., $f_c$ and $2f_c$; the inner ring rotating frequency $f_i$; and the combination frequencies, namely $f_i$-$2f_c$, are deeply affected by the guiding clearances. The beat vibration of the cage motion when guiding clearance is relatively small appears in the time domain.

As an inner-ring-guided bearing, the inner ring rotating frequency $f_i$ dominates in the spectrum of the cage motion, and two times cage rotating frequency takes the second place, when guiding clearance is at $C_g$ = 0.1 mm. The smaller the guiding clearance, the easier it is to contact the cage and to transmit the inner ring motion to the cage due to the impact, so the dominant frequency component is $f_i$.

In the operating condition of $C_g$ = 0.2 mm, the amplitude of $f_i$ decreases, while that of $2f_c$ increases and becomes the dominant frequency component. This is because larger guiding clearance will increase the free space between the cage-guiding face and the inner ring and impact with the ball. This will weaken the impact of the inner ring; it is easy to bring out smaller contact force applied to the cage from the inner ring but a greater one from balls.

As the guiding clearance increases, the impact of cage with balls and inner ring will gradually decrease and the unbalance forces will play a more important role, so the regular cage motion with the main frequency component $f_c$ will appear.

### 3.2. Wear of the Bearing Cage

The wear of the cage-guiding surfaces from the disassembled test bearing after the same operating with the three different guiding clearances cycles is shown in Figure 6. It can be seen that the degrees of cage wear are significantly different through observation of wear width and color on the guiding surfaces. The smaller the guiding clearance is, the more serious the wear of the guiding surface, as shown in Figure 6.

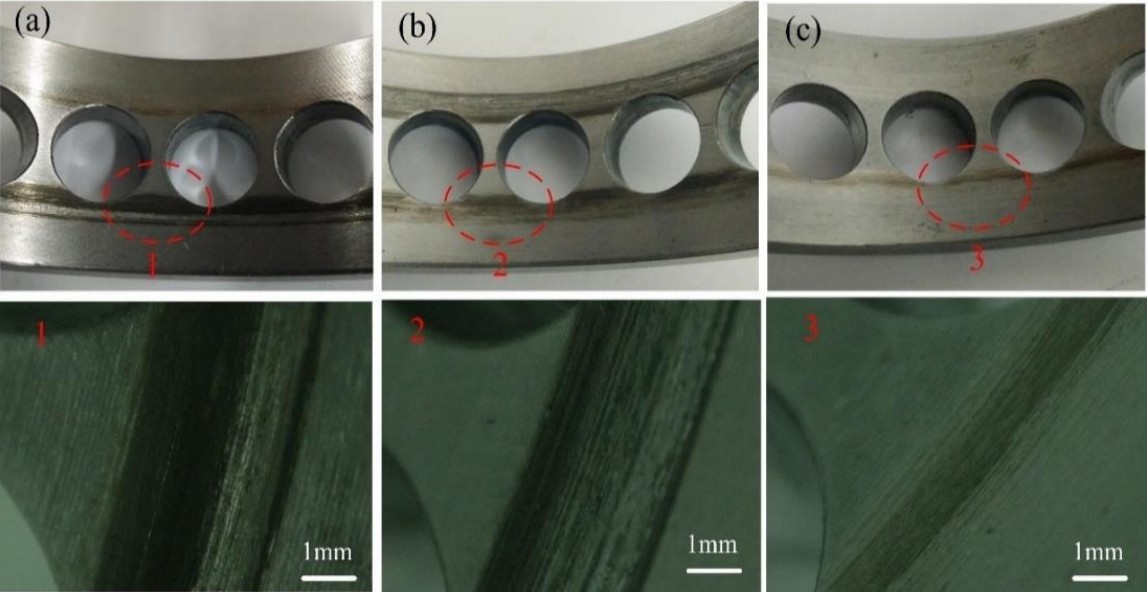

**Figure 6.** Wear of cage guiding surface with different guiding clearances. (**a**) $C_g$ = 0.1 mm, (**b**) $C_g$ = 0.2 mm, (**c**) $C_g$ = 0.4 mm.

Compared with the cage motion spectra and trajectories in Figure 5, it can be found that when the cage motion irregularly, especially when $f_i$ takes a high proportion because of the frequently and greater impact transmitted from the inner ring to the cage due to higher contact forces with small clearances, the wear of the guiding surface is serious, where the wear width is 3.5 mm in Figure 6a,b.

When the guiding clearance is large enough, the regular cage motion with the main component $f_c$ will appear, because larger guiding clearance will weaken the impact between the cage and the inner ring, and the cage unbalance forces will play a more and more important role. Accordingly, it is nothing serious for the cage guiding surface with larger clearance in which the wear width is 2 mm in Figure 6c.

Figure 7 shows the wear of cage pockets with different guiding clearance. It is obvious that the wear of the cage pocket in Figure 7b is the most serious with obvious black band marks, while the wear is not obvious in Figure 7a,c.

By comparing with the cage motion spectrum in Figure 5, it can be found that $2f_c$ is the main frequency in cage motion when $C_g$ = 0.2 mm, because the intermittent and recurring rub-impact between the cage pocket and balls is severe, which aggravates the wear.

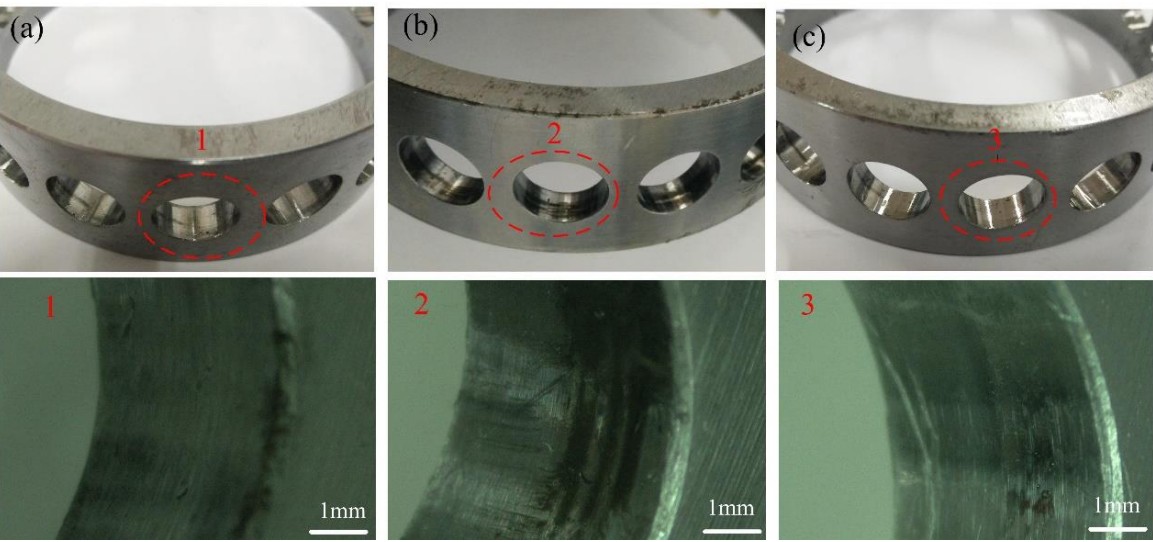

**Figure 7.** Wear of cage pocket with different guiding clearances. (**a**) $C_g$ = 0.1 mm, (**b**) $C_g$ = 0.2 mm, (**c**) $C_g$ = 0.4 mm.

## 4. Effect of Cage Pocket Clearance

### 4.1. Radial Motions of the Bearing Cage

The vibration waveforms, spectra of the cage motion in the vertical direction, and normalized trajectories of the cage with three different cage pocket clearances at the rotating speed 6000 rpm and constant preload (axial load 1000 N and radial load 500 N) and same guiding clearance 0.45 mm are illustrated in Figure 8.

It is can be seen from Figure 8 that the cage radial motion is always periodic with main frequency components $f_c$, $2f_c$, and $f_i$. With the increment of cage pocket clearance, the whirl trajectory of cage are gradually transformed from well-defined patterns to complicated ones, but it does not cause obvious instability of the cage, which is consistent with the simulations in reference [5].

The cage with small pocket clearance ($C_p$ = 0.1 mm) moves in a stable state, and its whirl trajectory is regular. When the pocket clearance reaches 0.25 mm and more, the amplitudes of $f_c$ and $f_i$ decreases, but a random broadband near $f_c$ appears in the spectrum of its cage motion. Because larger pocket clearance gives more free space for the rolling balls, the impact of the cage and balls will be undermined, and more space for the cage independent of the inner ring and balls will be obtained, which makes the cage motion pattern more irregular.

### 4.2. Wear of the Bearing Cage

The wear of cage-guiding surfaces and pockets from the disassembled test bearing after the same operating cycles with the three different pocket clearances is shown in Figures 9 and 10. The degrees of cage wear can be evaluated by use of wear width and color. It can be clearly seen that the degrees of the guiding surfaces' wear are significantly different. With the pocket clearance increases, the wear of the guiding surface becomes more serious, corresponding to the amplitudes of $f_i$ shown in Figure 8, and then the wear of the pocket is also decreased, but the distinguishing features is smaller than the guiding surface, as shown in Figures 9 and 10. Shrinking the pocket clearance leading to the small free space of ball will impel continued impact between the cage and balls, and then push the cage towards the inner ring, which will lead to severe wear of the cage pocket and the guiding surface. Increasing the pocket clearance will weaken the impact between the cage and balls.

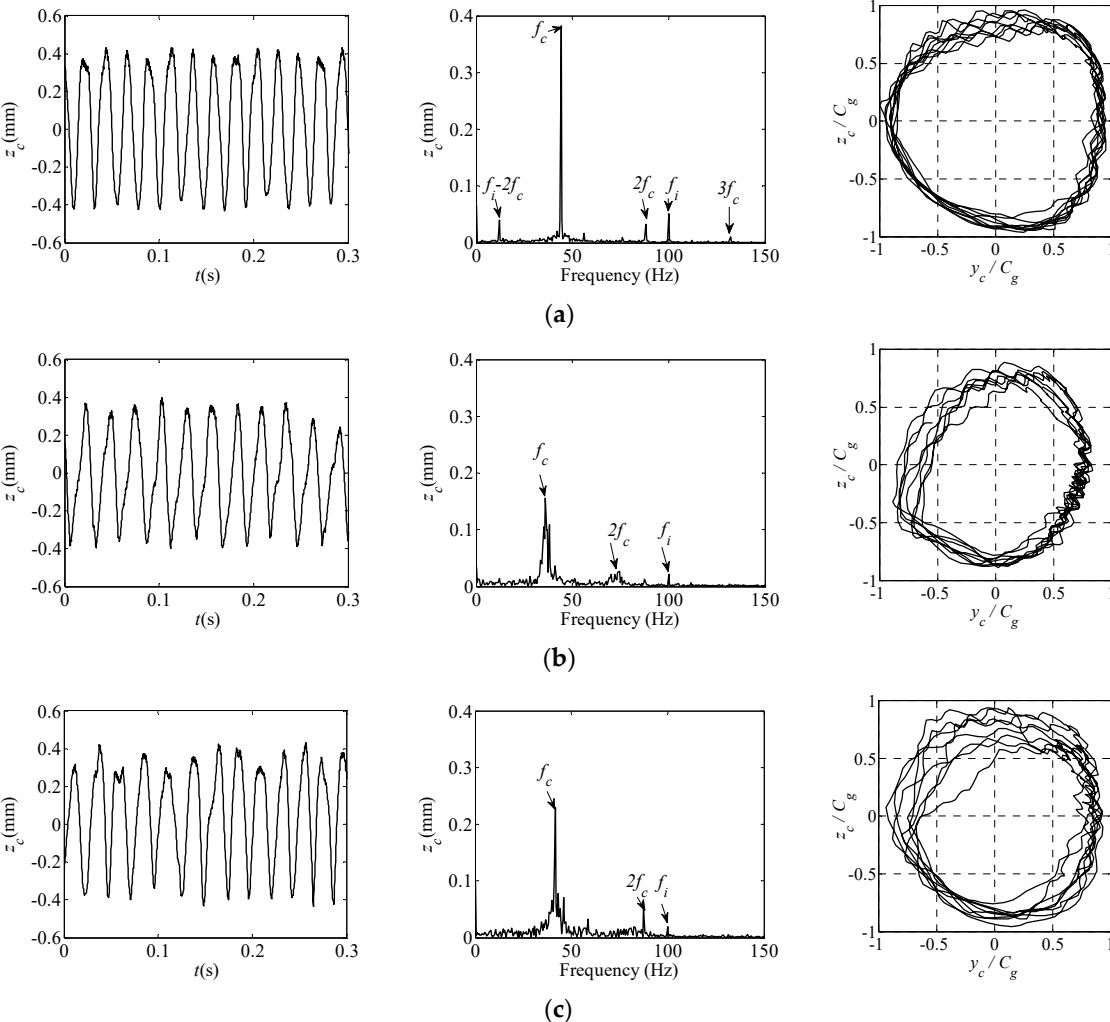

**Figure 8.** Radial motions of the bearing cage with different cage pocket clearances ($\omega_i$ = 6000 rpm). (**a**) Waveform, spectra and normalized trajectory of cage motion when $C_p$ = 0.1 mm; (**b**) waveform, spectra and normalized trajectory of cage motion when $C_p$ = 0.25 mm; (**c**) waveform, spectra and normalized trajectory of cage motion when $C_p$ = 0.4 mm.

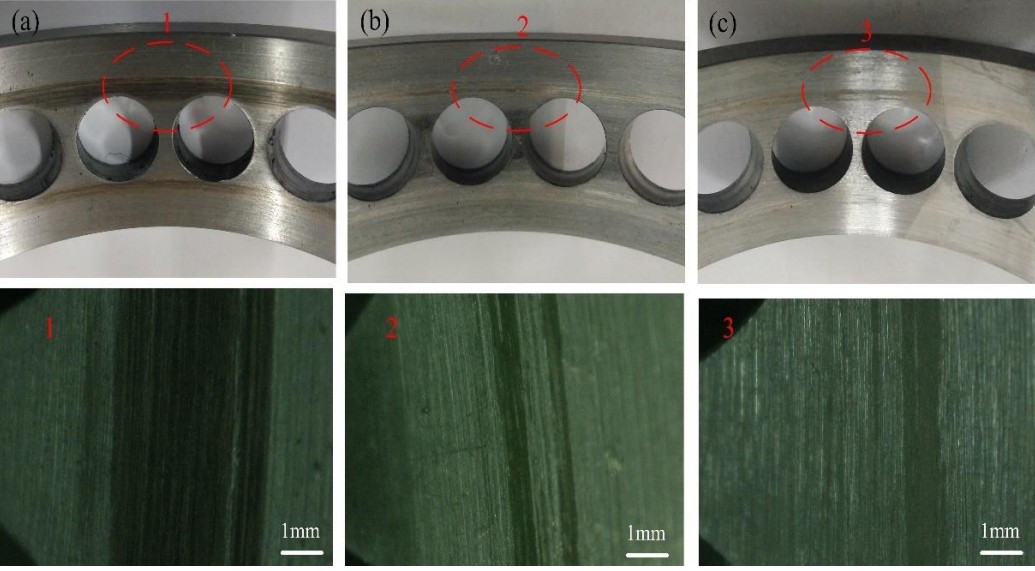

**Figure 9.** Wear of guiding surface with different pocket clearances. (**a**) $C_p$ = 0.1 mm, (**b**) $C_p$ = 0.25 mm, (**c**) $C_p$ = 0.4 mm.

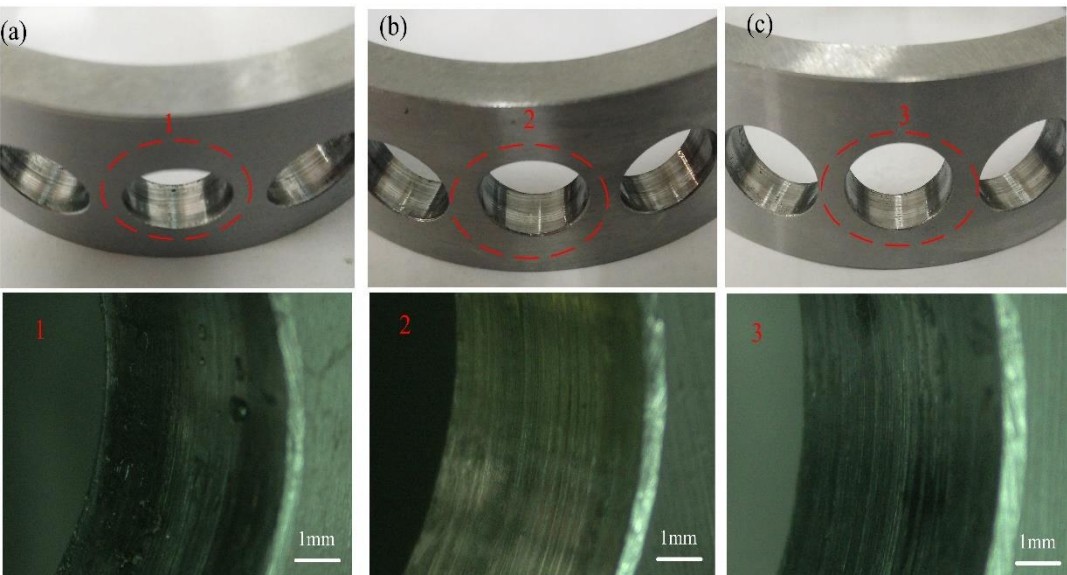

**Figure 10.** Wear of pocket with different pocket clearances. (**a**) $C_p$ = 0.1 mm, (**b**) $C_p$ = 0.25 mm, (**c**) $C_p$ = 0.4 mm.

## 5. Conclusions

In this paper, the cage motion and wear in a ball bearing affected by its clearances have been experimentally investigated on a test rig with a redesigned cage, and the main conclusions are as follows:

(1) Both the guiding clearance and pocket clearance of the bearing cage have significant effects on its motions and wear, and the guiding clearance has a more obvious effect.

(2) With the increment of cage-guiding clearance, the amplitudes of the main frequency components (including the cage rotating frequency and its two times, i.e., $f_c$ and $2f_c$, the inner ring rotating frequency $f_i$) will significantly change in the time domain and the trajectories of the cage will gradually change from irregular to regular. However, when increasing cage pocket clearance, the whirl trajectories change from well-defined patterns to complicated ones, which will further cause instability of the cage.

(3) Additionally, the bearing cage wear is closely related to the its motion. If the inner ring rotating frequency $f_i$ dominates in cage motion, the cage-guiding surface will wear seriously due to continual impact between the cage surface and the inner ring with smaller guiding clearance. While cage motion is dominated by two times cage frequency $2f_c$ in spectrum domain, the cage pocket will wear more seriously due to contact and rub-impact between the cage pockets and balls.

**Author Contributions:** Methodology and investigation, B.W.; original draft preparation, M.W.; writing—review and editing, X.Z.; formal analysis and data curation, J.Z.; funding acquisition, W.S. All authors have read and agreed to the published version of the manuscript.

**Funding:** This work was financially supported by the financial supports from National Key R&D Program of China (No. 2020YFB2006801), the National Natural Science Foundation of China (No. 51905069), the Department of Education of Liaoning province (No. LJKZ0536 and LJKZ0542), the Natural Science Foundation of Liaoning Province (No. 2019-ZD-0088), and the High-level Talents of Dalian City (No. 2018RQ18).

**Conflicts of Interest:** The authors declared no potential conflict of interest with respect to the research, authorship, and/or publication of this article.

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
