# Peer review of "Experimental Investigation of Clearance Influences on Cage Motion and Wear in Ball Bearings"

_applsci, doi:10.3390/app112411848_

Round 1

Reviewer 1 Report

You will found my comments in the attachment.

Author Response

Thanks for your valuable recommendation. We have revised and improved the manuscript carefully according to the following suggestions. The revised and re-written parts are marked in red color in the text. Thank you very much for your work for improving our manuscript.

Our responses to your comments are as following.

Reviewer 2 Report

The results are interesting, and the conclusions are well supported by the analysis of the results. However, the section describing the test ring is very condensed even though it refers to quote 17. 
Did you use the same test bench? Did you do any signal processing?

I suggest expanding the methodology. Specifically, the test bench part since the experiment design has a direct impact on the results.

Author Response

Thanks for your valuable recommendation. We have revised and improved the manuscript carefully according to the following suggestions. The revised and re-written parts are marked in red color in the text. Thank you very much for your work for improving our manuscript.

Round 2

Reviewer 1 Report

The authors have taken into considerations all our comments and subsequently have improved the manuscript. It is now apt for publication, save for some English corrections required.